# Artificial Neural Networks Performance in WIG20 Index Options Pricing

**DOI:** 10.3390/e24010035

**Published:** 2021-12-24

**Authors:** Maciej Wysocki, Robert Ślepaczuk

**Affiliations:** 1Quantitative Finance Research Group, Faculty of Economic Sciences, University of Warsaw, Ul. Długa 44/50, 00-241 Warsaw, Poland; maciej.wysocki13@gmail.com; 2Quantitative Finance Research Group, Department of Quantitative Finance, Faculty of Economic Sciences, University of Warsaw, Ul. Długa 44/50, 00-241 Warsaw, Poland

**Keywords:** option pricing, artificial neural networks, implied volatility, supervised learning, index options, Black–Scholes–Merton model

## Abstract

In this paper, the performance of artificial neural networks in option pricing was analyzed and compared with the results obtained from the Black–Scholes–Merton model, based on the historical volatility. The results were compared based on various error metrics calculated separately between three moneyness ratios. The market data-driven approach was taken to train and test the neural network on the real-world options data from 2009 to 2019, quoted on the Warsaw Stock Exchange. The artificial neural network did not provide more accurate option prices, even though its hyperparameters were properly tuned. The Black–Scholes–Merton model turned out to be more precise and robust to various market conditions. In addition, the bias of the forecasts obtained from the neural network differed significantly between moneyness states. This study provides an initial insight into the application of deep learning methods to pricing options in emerging markets with low liquidity and high volatility.

## 1. Introduction

The history of neural networks (NNs) started in the early 1940s, when McCulloch and Pitts [1] proposed the first computational model for NNs. Throughout the years, many upgrades and improvements have been proposed. The popularity of NNs started to grow in 1974, when Werbos [2] published his work about the backpropagation algorithm that enabled the operational training of models. Machine learning (ML) techniques have been broadly used in finance in many different applications, such as forecasting stock price movements, pricing derivatives, the preventing credit frauds. NNs were applied for a variety of tasks, such as algorithmic trading, modeling volatility, or speeding up processes of calibrating parametric models.

Although the performance of NNs has already been described in different papers, most of them focused on simulated markets or data from the New York Stock Exchange, with the approach of boosting the performance of the Black–Scholes–Merton model (BSM). The main aims of this paper were the exploration of deep learning possibilities in option pricing and the analysis of the market data-driven approach for NNs training for a developing market. None of the previous works have covered the topic of the machine learning (ML) approach to pricing derivatives on emerging markets with relatively low liquidity and high volatility. Considering previous results of NNs, these models might turn out to be a solution for problems occurring in pricing contracts quoted on emerging markets. The goal of this research was to design the proper architecture of the NN for a data-driven approach and to test its performance in comparison with the traditional BSM model.

The first research hypothesis was that neural networks trained on real-world market data could perform better than the Black–Scholes–Merton model in terms of pricing errors. Similar to the standard machine learning development process, common evaluation metrics were compared for both NN and BSM models to decide which of these two gave more accurate prices. In order to verify the hypotheses and provide the most robust ML model, the hyperparameters tuning for NN were conducted.

The assumption that NNs might, in fact, perform at least as well as the traditional BSM model was based on many previous kinds of research, such as [3,4]. As these articles stated, properly designed NNs trained on market data can significantly outperform other models, including BSM. 

The second hypothesis was that one could observe a difference in pricing errors of the NN considering the moneyness ratio. For different moneyness states, there could be various error distributions. The revealing pattern in error distribution might provide an explanation of its magnitude. Moreover, for purposes of future NN architectures, such analysis might be helpful when deciding whether options with particular characteristics should be excluded from the training process.

Considering the properties of options and the differences between moneyness states, it is a reasonable assumption that the pricing error will differ for every possibility. Kokoszczyński [5] provided evidence from the Polish market that, indeed, different moneyness ratios and time to maturities reveal patterns of the pricing error distribution. The same has been carried out for the Japanese option market, regardless of the chosen model and its characteristics [6]. 

The rest of the paper is organized as follows. Firstly, a literature review is provided, followed by a chapter consisting of the option pricing models included in the paper and a methodology description with an introduction to NNs. The third chapter is devoted to components of ANNs, as well as a description of proper network architecture development. Subsequently, data description and its preprocessing for purposes of ANN fitting are provided. The next chapter presents empirical results, model performance, and comparison. Verification of main hypotheses, summary, and proposals for further research are placed in the conclusion of this paper.

## 2. Literature Review

The paper entitled “The Pricing of Options Corporate Liabilities” published by F. Black and M. Scholes [7], together with the paper “Theory of Rational Option Pricing”, written by R. Merton [8], introduced the BSM model, providing a simple framework for practical implementations, which gave reliable results. However, many of its assumptions were quickly proven to be non-realistic.

Bates [9] provided a very exhaustive discussion of previous empirical research concerning option pricing, especially the BSM model. The work of Bakshi et al. [10] revealed inconsistency of the BSM model between different moneyness and maturities. Moreover, it was shown that introducing stochastic volatility [11,12] and stochastic volatility jumps [13] does indeed improve the performance of option pricing models. 

Implied volatility was said to have huge predictive power in forecasting the future volatility on the markets to address issues with biased volatility indicators. However, the work of Canina and Figlewski [14] clearly stated that the implied volatility calculated with the BSM model has practically no correlation with future volatility. Similar work of Fleming [15] showed that implied volatility was a biased estimator; however, it did contain valuable information about the future realized volatility. Although it was proven to be biased, the implied volatility has an advantage over other estimators when used in the BSM model, as shown in many previous works [16,17]

Park et al. [18] provided a comparison of a few ML algorithms—support vector regression (SVR), Gaussian process regression (GPR), and ANN with parametric methods—the BS model, the Heston model, and the Merton model. It turned out that ML methods significantly outperformed the BS model, as well as performed comparably well to other parametric models, depending on moneyness and maturity. Similar conclusions were stated by Wang [19] in an article that summarized the performance of SVR in currency options pricing. Results of ML methods obtained on Hong Kong Derivatives Market [20] also confirmed that the use of SVR and NN improved pricing accuracy and provided better, more reliable results. 

The very first papers concerning the use of ANNs for purposes of derivatives pricing were the articles of Malliaris and Salchenberger [21] and Hutchinson et al. [22]. The results from both papers were promising, as in the first work the ANNs managed to outperform the BSM model. The second article investigated the hedging performance of the NNs. 

The simplest approach to pricing options with NNs was feeding the network with the price of the underlying asset and strike price without any processing along with the other parameters used in the BSM model; this can be found in [23,24,25]. Nevertheless, this approach was far less popular than the use of transformed spot price and strike price. The most common transformation is a division of the underlying asset price by the strike price. Such data preprocessing can be found in [26,27,28,29].

Yang et al. [4] proposed the use of the gated NNs that not only provide reliable prices of the options, but also contain a guarantee of economically reasonable and rational results. When it comes to portfolio hedging, the long short-term memory (LSTM) RNN was said to be outperforming conventional methods [30].

Another approach to leverage the flexibility of deep learning methods was presented in [31], where the author proposed a two-step approach, also called a stack of models. The options were first priced using three different classical methods and these estimations were then taken as an input to the ANN. The author was able to deliver evidence for a superior pricing accuracy of such an approach over classical models.

## 3. Methodology and Option Pricing Models

### 3.1. Terminology and Metrics

An option is a contract giving the holder a right to purchase (a call option) or to sell (a put option) a fixed amount of underlying asset at a specific date—the expiration date. European style options are the ones that cannot be exercised before the day of expiration. 

Moneyness describes an actual profit for the owner of an option, from exercising the option immediately at that time. In-the-money (ITM) options are the ones that would have positive intrinsic value if they were exercised today. Similarly, out-of-the-money (OTM) options are the ones that would have negative intrinsic value if they were exercised today. At-the-money (ATM) options fill in the last possibility, which is that the current price and strike price are the same, so the theoretical profit is equal to zero. To classify options within their current moneyness, the moneyness ratio (MR) is used, expressed as follows [5]:(1)$$MR=\frac{S}{K\cdot e^{-r\cdot \tau }}$$
where S is the spot price of the underlying, K is the strike price, r is the risk-free rate, and τ is the time to maturity. For call options, moneyness ratio in the range [0; 0.95) refers to OTM options, the moneyness ratio in the range [0.95; 1.05) means the option is ATM, and options with MR higher than 1.05 are in-the-money. Put options are classified using the same ranges but in reverse order, so that the first range is for ITM options and the last one is for OTM options. 

When it comes to the model evaluation and comparison between different methods, error metrics must be introduced. Statistics used in this paper are typical for any work concerning regression problems [32]:Mean absolute error (MAE)
(2)$$MAE=\frac{1}{n}\overset{n}{\underset{i=1}{\sum }}\left|Y_{i}-{\hat{Y}}_{i}\right|$$

Mean square error (MSE)


(3)
$$MSE=\frac{1}{n}\overset{n}{\underset{i=1}{\sum }}{\left(Y_{i}-{\hat{Y}}_{i}\right)}^{2}$$


Root mean square error (RMSE)


(4)
$$RMSE=\sqrt{\frac{1}{n}\overset{n}{\underset{i=1}{\sum }}{\left(Y_{i}-{\hat{Y}}_{i}\right)}^{2}}$$


Mean absolute percentage error (MAPE)

(5)$$MAPE=\frac{1}{n}\overset{n}{\underset{i=1}{\sum }}\left|\frac{Y_{i}-{\hat{Y}}_{i}}{Y_{i}}\right|$$
For each case $Y_{i}$ is the real value and ${\hat{Y}}_{i}$ is the value calculated by the model.

### 3.2. Black–Scholes–Merton Model

The model used in this paper is the framework BSM model with the continuous dividend paid by the underlying asset. The model is based on the partial differential equation, namely the Black–Scholes Equation (7), as follows:(6)$$\frac{\partial V}{\partial t}+\frac{1}{2}\sigma^{2}S^{2}\frac{\partial^{2}V}{\partial S^{2}}+rS\frac{\partial V}{\partial S}-rV=0$$
where *V* is the price of the option at time *t*, *S* is the price of the underlying at time *t*, *r* is the risk-free interest rate, and *σ* is the standard deviation of the underlying asset’s returns. The solution that is the price of either put or call price given by the model is sought to satisfy Equation (6). The prices of European options can be obtained using the following Formula (7):(7)$$P_{c}\left(S_{t},\tau \right)=S_{t}\cdot e^{-q\cdot \tau }\cdot N\left(d_{1}\right)-K\cdot e^{-r\cdot \tau }\cdot N\left(d_{2}\right)$$
(8)$$P_{P}\left(S_{t},\tau \right)=K\cdot e^{-r\cdot \tau }\cdot N\left(-d_{2}\right)-S_{t}\cdot e^{-q\cdot \tau }\cdot N\left(-d_{1}\right)$$
(9)$$d_{1}=\frac{lnln\left(\frac{S_{t}}{K}\right)+\left(r-q\right)\cdot \tau }{\sigma \sqrt{\tau }}+\frac{\sigma \sqrt{\tau }}{2}$$
(10)$$d_{2}=\frac{ln\;ln\left(\frac{S_{t}}{K}\right)+\left(r-q\right)\cdot \tau }{\sigma \sqrt{\tau }}-\frac{\sigma \sqrt{\tau }}{2}=d_{1}-\sigma \sqrt{\tau }$$
where $P_{C}$ is the price of the call option and $P_{P}$ is the price of the put option. In these equations, an additional symbol, q, denotes the dividend rate, K stands for the strike price, and $N\left(\cdot \right)$ is the cumulative distribution function of the standard normal distribution. 

The one parameter that cannot be directly observed on the market is the volatility of the underlying asset. In this paper the volatility estimator is based on the historical returns from the underlying and calculated using the following formula:(11)$$HV_{n}=\sqrt{T}\sqrt{\frac{1}{n-1}\overset{n}{\underset{t=0}{\sum }}{\left(u_{t}-\bar{u}\right)}^{2}}$$
(12)$$u_{t}=ln\frac{S_{t}}{S_{t-1}}$$
where $u_{t}$ is a logarithmic rate of return from the underlying asset, u is the mean of all logarithmic returns in the sample, T is the number of trading days in a year, and n is the size of the sample. Typically, T is equal to 252 trading days in the year. The memory of the volatility process is chosen to be 60 days.

There are many other approaches to modeling volatility that could be used to obtain estimators introduced to the BSM formulas, such as realized volatility [33], stochastic volatility [11,12], or implied volatility, calculated from observed market options prices [5,6,34]. An executive review of various volatility estimators can be found in [35].

### 3.3. Artificial Neural Network

#### 3.3.1. Architecture of Artificial Neural Networks 

The class of NNs used in this paper is the multilayer perceptron (MLP) [36,37], also called the feedforward network, which consists of at least three layers of neurons: the input layer, the output layer, and at least one hidden layer between them. There are no restrictions for the number of layers or neurons in them, so one must find a proper architecture depending on the data the NN is fed with. The very first layer of MLP is the input layer, which consists of a data vector and a bias term from which the weighted sum is calculated and then fed forward to the first hidden layer. In each layer of the network, there is a chosen activation function that is responsible for the activation of the layer if the output from the previous one exceeds the threshold of activation. The output of the activation function from each neuron is then passed further to the next layer of nodes, where the weighted sum is calculated again. This process continues until the output layer is introduced where the output vector is calculated. 

The basic component of ANNs is a neuron, also called a node. The neurons take the input either from the initial data set or from the previous layer and combine it with an optimal threshold, calculated using the activation function. These functions’ task is introducing non-linearity to the model, as well as creating a differentiable transition as the input changes. The neurons are attached to connections that are responsible for assigning weights, in such a way that more valuable information has a higher weight. Between layers, a full connection can exist, and a dropout could be introduced. Dropout is a technique used for preventing the ANN from overfitting to the training dataset and it consists of dropping out randomly selected nodes from a single layer.

Training an NN for regression problems relies on training the model on the example of input and output pairs from the training dataset. The training process is conducted by minimizing the differences between real and predicted values to maximize the accuracy of the fitted values. The errors are expressed using a cost function which depends on the type of the problem. The cost function is evaluated in every run and then the weights in connections between the neurons are updated to optimize the function. Typically, the learning process continues as long as the error is reduced; therefore, intentionally, the cost function reaches its global extremum.

#### 3.3.2. Backpropagation and Optimization

Backpropagation [2] is an algorithm used in training MLPs for supervised learning problems. This method is designed to adjust the weights in a connection between neurons in the way that the cost function is minimized. The algorithm calculates the loss function’s gradient with respect to each weight during the training process. Starting from the output layer, partial derivatives are calculated through every layer to the input layer and then for each of them, the algorithm returns gradient with respect to adequate weights. The main advantage of backpropagation is its efficiency, which allows the use of gradient-based optimization techniques. Due to the backpropagation algorithm, training the NN can be conducted as an iterative process of updating weights. 

The gradient-based optimization techniques are designed to search for global optima of a function in directions pointed by the gradients at the specific point. The most common technique is the gradient descent algorithm used for finding a local minimum of a differentiable function. Nevertheless, there are many alternatives that could be used for optimization purposes, such as stochastic gradient descent (SGD) or adaptive moment estimation (ADAM), in which both the gradients and the second moment of the gradients are calculated and used for weights updating [38].

#### 3.3.3. Hyperparameters

The hyperparameters are a type of parameter that are arbitrarily set before the learning process starts and do not change through the training phase. As the option pricing is a supervised regression problem, the chosen loss function to be minimized was MAE. 

The basic hyperparameters are the number of layers in an NN and the number of neurons in each layer. Both must be defined at the beginning of architecture design. Typically, NNs used for purposes of option pricing do not have too many layers [39] and a search for the optimal number should be performed, rather using the trial-and-error method [40]. Following this technique, the proper number of layers was found to be six, including the input and the output layers. The search for the optimal number of layers started from one hidden layer and, consequently, networks up to six hidden layers were tested. 

For the first stage of the NN architecture design, the number of nodes was chosen along with the activation functions, a batch size, number of epochs, a dropout rate, and an optimizer. At that point, the task was to find an initial set of parameters that were performing well and were stable to use as a starting point in the tuning phase. The final values of parameters were chosen in the process of hyperparameters tuning conducted for all the following hyperparameters: batch size, dropout rate, optimizer, initializer, learning rate, β1, and β2. 

The batch size is the parameter defining how many examples are used in one backward pass of the learning process. It defines how many samples from the training data are used in one complete training pass through the network. 

The dropout rate refers to ignoring randomly selected nodes during one pass of the training phase in a certain layer. The dropouts are introduced for chosen layers to prevent the algorithm from overfitting to the training data. 

The optimizers are algorithms responsible for updating the attributes of NNs, such as the weights and the learning rate, to reduce the error and minimize the loss function. 

The initializers are functions, defining the way to set the initial random weights for connections in each dense layer. Choosing a proper initialization technique is crucial, as only proper weights allow optimizing the function in a rational amount of time. When the initial weights are set incorrectly, the convergence to the global minimum is impossible. 

The learning rate is responsible for controlling how much the weights are updated in response to the evaluated cost function at every pass in the training process. Additionally, in the case of the Adam optimizer, instead of adapting the learning rate based on the average first moment, the average of the second moment of the gradients is used. The algorithm calculates an exponential moving average of the gradient and the squared gradient, and the parameters β1 and β1 control the decay rates of these moving averages.

The first stage of the NN architecture design was conducted as an iterative process of training the NN with another combination of hyperparameters, until all possible sets of the hyperparameters indicated in Table 1 were checked (5×6×1×4×2×4=960). The training was performed on 80% of the total dataset. Then, the results were summarized and the set that resulted in the lowest value of loss function was chosen. This approach allowed us to design an NN that gave stable and satisfactory results for that moment.

The hyperparameters tuning, performed on the same 80% of the total dataset, was conducted in the following way. For each of the parameters a set of possible values was chosen and then the network with framework architecture (Table 2) was trained using different values of just one parameter, with other parameters set to be constant. Such an approach allowed comparison between different hyperparameter values. Moreover, changing only one hyperparameter at a time ensured that the changes in the results were caused by the investigated parameter and not by the others.

#### 3.3.4. Results of the Hyperparameters Tuning

The number of epochs was set to 5 so that the algorithm would be responsible for updating weights 5 times during a single training process. The change in the number of epochs from 15 to 5 was done after careful analysis of the initial learning process and its error estimates. Moreover, reducing the number of epochs enabled the process of tuning of parameters to substantially speed up. Table 3 contains all the ranges or possible options for different parameters of the model. The activation function was not included in the tuning process because none of the other activation functions, besides ReLU, enabled the gradient to converge.

The number of neurons was chosen to be the same for every layer. Although the initial value of nodes was selected from a similar range, it was checked for different values once again on the framework to confirm the results.

As clearly visible in Table 4, values were very similar in each case. The final number of neurons at each layer was chosen to be 500, since, for that number, the cost function was monotonically decreasing during the training process in contrast to other possible numbers of neurons, for which either MAE or MSE were behaving in a non-monotonic way.

When it comes to the batch size (Table 5), the optimal value was different from the one chosen in the first stage. In the final model there are 1000 examples of input and output data during one backward pass in the training process. Although the final values of MAE and MSE were slightly higher than for the batch size set to 2000, the learning process for the batch size equal to 1000 is more stable; therefore, this value was chosen.

The dropout rate was set to 0.2, because for that value, the process remained the most stable among the others as well as the error metrics evaluated after the first epoch was the lowest (Table 6). The stability of the process means that no sudden jumps upward or downward of the error metrics were observed during the training process.

As the Table 7 presented, the worst results were obtained for Adadelta and SGD. Better results were obtained using Nadam and the best results were obtained for Adam, Adamax, and Adagrad. Since Adam provided reproducible results and it was the most popular one in financial applications, it was chosen as the final optimizer.

The highest MAE and MSE values were obtained for random and random uniform normal initializing function (Table 8). Glorot normal and lecun uniform performed very similarly. The final method of random weights assignment was chosen to be lecun normal, due to its monotonically decreasing error metrics.

For the analysis of learning rate and β1 and β2 hyperparameters, another approach was taken. These parameters were investigated together due to their similarity and the roles that they have. All of them were parameters of the optimizer that influenced the model flexibility—that is, they influenced how much the model was updated in every pass of the training phase. Table 9 summarizes the best 5 runs out of 100. The parameters chosen to be in the final architecture of the model are the learning rate at the level of 0.001, β1 equal to 0.9, and β2 equal to 0.9999. 

The number of epochs was chosen to be 15, so the learning process took 15 repeats of passing the entire set backward and forward through the algorithm. Different values of the epoch were checked to find the number of epochs that allowed the network to converge. The aim was to obtain a stable process of learning without visible overfitting. As Figure 1 shows, the learning process remained stable without any unexpected jumps or random disruptions. The loss function stabilized with a final value around 0.02321 and the MSE metric stabilized near 0.0025. The NN learned fast as the process started to remain stable at the 5th epoch and then the loss function remained at similar values.

## 4. Data Description

The dataset used in the research was gathered from stooq.com (accessed on 3 June 2020) and Warsaw Stock Exchange (WSE) and its core parts were daily quotes of WIG20 index European options and WIG20 index from the WSE.

### 4.1. Data Distribution

The quotes covered the period from the beginning of January 2009 to the end of November 2019. Such a wide time frame of the data allowed many different conditions on the market to be covered. The 3 month Warsaw Interbank Offer Rate (WIBOR3M) was used as an estimator of the risk-free interest rate, similarly to that used in [5]. The options included in the dataset were all of these quoted on the WSE, so the research covered many different strike prices and maturities. For the modeling purposes, the time to maturity was calculated in years, where one year was 252 trading days. As a dividend rate estimator, a continuous dividend yield from the WIG20 index was used. The historical volatility estimator was calculated using the method described in one of the previous chapters.

For modeling purposes, the close price was assumed to be the proper option’s price and was then used as a true value. A number of the records in the collected dataset was equal to 139,371, where 68,285 observations concerned the call options, and the remaining 71,086 observations concerned the put options. The dataset was then well balanced and a large amount of data for both put and call options was included. Table 10 presents a summary of the descriptive statistics for the whole dataset

The options prices were distributed in a very wide range between 0 and 1600. The mean price was near 60 and the median price was 21, so the distribution was uneven and outliers were probably introduced to the dataset. As seen in Figure 2, the intuition concerning the outliers in the dataset was confirmed. Prices close to 0 dominated the dataset; however, there were some observations with prices higher than 100.

Table 11 summarizes descriptive statistics for put and call options distinguished between three moneyness states. As is clearly visible, most of the observations were OTM options. There were only around 7100 ITM calls and the same amount of ITM puts. For both types, there were 25,000 observations ATM both calls and puts. Moreover, the prices were the highest for ITM options, while both OTM and ATM were cheaper, as all the statistics were smaller for them.

### 4.2. Data Preprocessing for Neural Network

For purposes of modeling with an NN the dataset first had to be divided into two subsets. The first, larger one is called the training or in-sample set, and it was used to train the model. The second, smaller one is called the testing or out-of-sample set, and it was used for validation purposes. A test sample was used to verify whether introducing the model to the new data will result in at least comparable results, as well as check if there was overfitting to the training sample. In this paper, 80% of the initial data was used as the training set and the remaining 20% was used as testing data. There is no golden rule specifying how to divide the data and the discussion on this topic continues [41]. Some authors have said that the time series characteristic of the data should not be interfered with, while others claim that the ability to catch different market conditions was more important. In this paper the second approach was taken, and the data was split with respect to varying price distribution. This means that the intent was to feature the training data with as many different market conditions as possible. 

The main aim of preprocessing was to change the distribution and range of the data. Applying statistical models, such as NNs, typically requires such preparations with respect to the characteristics and the abilities of the models. When it comes to the application of NNs in option pricing, the typical transformation is normalizing the data [42] in the following way:(13)$$X_{preProcessed}=\frac{X-\bar{X}}{\sqrt{Var\left(X\right)}}$$
where X denotes a vector of values for a single variable in the dataset and $\bar{X}$ denotes the mean value. The same transformation has been used in this research in order to obtain reliable and unbiased results by methods widely described in other papers. 

The last issue, when it comes to modeling with the use of NNs, was choosing which input the model should be fed with. One option was to feed the network with just spot price, strike price, and time to maturity [43,44]. A more common approach was to use option price, divided by strike price and time to maturity [18,45]. The most common approach found in the literature is using spot price divided by strike price along with time to maturity, interest rate, and volatility [24,31,46]. As this paper aimed to compare the performance of a BSM model and NNs in pricing options, the ML model was chosen to be introduced to the same data as that used in the BSM model. In other words, the input variables were spot price, strike price, interest rate, continuous dividend rate, time to maturity, and volatility. Moreover, the spot price was divided by the strike price. Similarly, the output of the NN was chosen to be the option’s price divided by its strike price.

## 5. Empirical Results

### 5.1. Cross-Validation Results

The k-fold cross-validation was conducted to check if the overfitting was introduced to the model as well as verify its abilities without using the out-of-sample dataset. This method consists of randomly dividing the data set into k folds of equal size that are then used to train and test the model [32]. The first group is left as out-of-sample data and the remaining k-1 groups are used to train the model that is then validated on the data left and the error metric is calculated. This procedure is then repeated k times and, as a result, there are k estimates of test error that are then averaged and treated as an out-of-sample error that could be compared with in-sample errors. 

The number of folds that the in-sample data was divided into was set to 5. The NN was trained 5 times on slightly different datasets, and each time, it estimated on the remaining part of data. The results below summarize the errors calculated after validation on a single fold.

The main conclusion from Table 12 was that no overfitting was detected; therefore, it was confirmed that the model was designed correctly. The performance was very stable, and errors remained low. It could be used to obtain the prices, without worries about bias resulting from improper training; in addition, there is no need to redesign the architecture of the NN or repeat the process of the hyperparameters tuning.

### 5.2. In-Sample Results

The in-sample results were the results obtained on the dataset (80% of all data) that was used to train the neural network. The BSM model was the benchmark model, so its results were summarized in the first order. There is no need for splitting the dataset when using the BSM model, as it does not require any training. Nevertheless, the results obtained with the BSM model were used as the first benchmark of the goodness of fit for the NN, as well as an indicator of the possible error range. Table 13 summarizes the error metrics for the prices obtained using the BSM model.

The first conclusion was that the quality of pricing with the use of the BSM model differed a lot between the types of options as well as their moneyness. To conduct a comparison of the pricing accuracy of the model, the mean average percentage error (MAPE) was used. ITM options were much more expensive compared with OTM options, which often had a price close to 0. MAPE allowed comparing the errors with the difference in the price already taken into consideration. Both call and put OTM options were priced with the highest percentage bias. For call options, the MAPE exceeded 1.22 when it came to OTM options, while for put options, this was close to 0.62. ATM options were priced with MAPE close to 0.37 for call options and 0.27 for put options. The ITM options were priced with the lowest percentage bias, close to 0.09 for both calls and puts.

The obvious conclusion from Table 14 was that the accuracy of the pricing was not stable between different moneyness states. The most reliable prices were obtained for the ITM options. ITM call options were priced with MAPE around 0.9, while the same metric for the OTM call options was close to 16.7. The OTM put options were priced with a mean average percentage error close to 10.5, while for the ITM options, this metric was near 0.89. The ATM call options were priced with the MAPE around 2.3, while this metric for the put options was closer to 1.91. 

When it comes to the comparison of the NN and the BSM model performance, the MAE was used to compare the accuracy of pricing for the same types of options in the same moneyness. Table 13 and Table 14 revealed that the BSM model provided much more reliable pricing than the NN. The OTM calls were priced by the deep learning model with MAE close to 22.8, and for puts, this metric was around 14.6. The same options were priced by the BSM model with MAE adequately 11.3 and 7.5. Similarly, for the ATM call options, the NN pricing resulted in MAE around 39.5 and for the put options around 44.8. The BSM model priced these options with the error metric adequately equal to 13.5 and 12.8. When it comes to ITM options, the parametric model priced the calls with an error metric of nearly 19.7, and the puts with the error metric were close to 24.3. The NN priced the ITM call options with MAE around 242.5 and the put options with MAE around 282.9. There was a huge difference between the methods and the NN performed much worse in pricing the ITM options. 

Although the training process remained stable and no overfitting was detected in the model, the resulting in-sample pricing performance was not satisfactory. So far, the prices provided by the NN were less reliable than those from the BSM model, as the resulting errors were higher for the deep learning model. Nevertheless, to verify the ML model, the out-of-sample results had to be compared.

### 5.3. Out-of-Sample Results

The final verification of ML methods, such as NNs, can be based on the out-of-sample results. For the out-of-sample results, the remaining 20% subset of the data was used. These data were not used before, so that the NN could be fed with a new input that consisted of 27873 observations. 

Table 15 shows that the error values varied depending on moneyness. Moreover, similar conclusions could be drawn as for the in-sample data. The ITM options were priced with the lowest MAPE, while the OTM options were priced with the highest error. The OTM call options were priced with MAPE near to 1.23, while the MAPE for the OTM put options was slightly above 0.6. The ATM options were priced with a similar MAPE, which was close to 0.36 for the call options and 0.3 for the put options. The MAPE metric was slightly above 0.09 for both ITM calls and ITM puts, which means that options in this moneyness state were priced the most accurately.

The obtained results (Table 16) suggested that no overfitting was introduced to the NN, because the out-of-sample metrics were comparable to in-sample ones. Moreover, the results were similarly biased and not any optimistic, so it was concluded that the data leakage was not a problem in this paper. The least reliable results, in terms of MAPE, were obtained for the OTM options—both the calls and the puts. For the OTM call options, the MAPE metric was above 17, while for the OTM put options, this error metric was equal to 10.5. The MAPE for the ATM options was equal 2.13 for the calls and 1.97 for the puts. For both ITM calls and puts the MAPE was close to 0.9.

Comparison of the results from Table 15 and Table 16 showed that the BSM model priced the options more accurately. The OTM call options were priced by the BSM model with MAE close to 11 and the put options with MAE close to 8. When it comes to the neural network, the error metric was adequately 23 and 14.7. The BSM model priced the ATM calls with MAE near to 13.8 and puts with MAE close to 13. The same options were priced by the NN with MAE adequately close to 40 and 44.75. The highest MAE was obtained for the ITM options, as the NN priced the calls with MAE nearly 246 and puts nearly 277.4. The same options were priced using the BSM model and the errors were close to 20 for the calls and nearly 23.9 for the puts.

Figure 3 compared the market price with the corresponding model price obtained from the BSM model (Figure 3a) and the NN (Figure 3b). The line y = x is a curve determining the perfect pricing where the model price and the market price are the same. For the BSM model, there were more dots below the curve, which means the model prices were positively biased. Therefore, the model tended to overprice options, which means that the prices obtained by the parametric model were higher than the corresponding real values. Nevertheless, the dots follow the straight line. This conclusion, however, could not be stated when it comes to the NN pricing. The non-parametric model tended to strongly underprice the options, which resulted in prices cumulated in the range between 0 and 75. The NN prices were often close to 10, while the real price of the option exceeded 1000. The OTM options that were priced with the lowest errors were the cheapest options. Even though the non-parametric model was trained using the market data, it was not able to catch the similarities to properly price options with varying moneyness and types.

### 5.4. Discussion of the Results

There could be a few reasons for such unsatisfactory results when it comes to modeling options prices with NNs. Firstly, the data used for the training purposes that was taken from the real-world market consisted mostly of the OTM options. Above 61,000 observations from the training sample were the OTM options, while only 11,401 observations were ITM options. Some authors suggest that filtering methods should be applied to the data to prevent such problems [26,44,47]. There are different methods that could be applied, such as selecting only ITM options or selecting only options that satisfied various maturity constraints. In this paper, the intention was to compare the pricing of options with parametric and non-parametric methods; therefore, no filtering was introduced. The BSM model was designed to provide prices that are more or less accurate, and the obtained results suggest that, although the errors varied for different characteristics, they were still comparable. When it comes to NNs, the resulting errors were different for various characteristics, and they were not comparable.

Secondly, the Polish Derivatives Market is an emerging market with typical problems for such markets, such as liquidity or non-synchronous trading [5]. This led to various price distributions on a wide range. As described in the previous section, the prices in the dataset varied between 0.01 and 1500 but were accumulated near 0. The prices above 1000 were so untypical that they could be treated as outliers, while the strike prices exceeded 1000. The NN turned out not to be robust for such a wide range of data. The main barrier was the lack of ITM options on the emerging markets with few participants. This was also a reason that filtering methods could not be applied, as the obtained data would not be reliable for the characteristics of the market.

## 6. Conclusions

The efficiency and accuracy of the parametric BSM model and the non-parametric NN in options pricing were verified using the data from Warsaw Stock Market. The NN was developed in a data-driven approach, which means it was designed and trained using real-world market data. The BSM model was used with the 60 days historical volatility estimator. To check the research hypotheses, the 10 years data was split into the training sample used for modeling purposes, and the testing sample was used for the validation of the model. The input to the NN consisted of the same variables as those used in the BSM model; however, the value of the WIG20 index as an underlying asset was divided by the strike price of an option. The output of the NN was the option price, divided by its strike price. The output was then transformed to analyze and compare error metrics for prices. The results were presented for call and put options, divided between three different moneyness states.

The results obtained from empirical analysis did not confirm the first research hypothesis, that the NNs trained on the market data are able to outperform the BSM model. In fact, the NN used for pricing options did not perform significantly better than the BSM model. The prices obtained using the data driven NN were far more biased than those from the parametric model. The NN was not robust to the varying market conditions, which are typical for emerging markets such as the WSE. 

The second hypothesis, that the prices provided with the NN are not robust to the varying moneyness states, could not be rejected. In terms of MAPE, the most accurate model prices were obtained for the ITM options, and the least accurate prices were obtained for the OTM options. The pattern in pricing accuracy was visible in the in-sample as well as the out-of-sample results.

To summarize, using NNs to price options for all maturities and moneyness states did not lead to a significant improvement in pricing accuracy. The BSM model was more robust to various market conditions and provided much more stable and reliable prices. One of the possibilities for dealing with the difficulties met by the NN would be in applying filtering methods to the data. Another possibility would be to develop different models for various characteristics, e.g., three NNs for three moneyness states. The results obtained by developing the NN on the wide dataset, consisting of options with different characteristics, were neither satisfactory nor reliable. 

This paper introduced an initial study of machine learning methods applied to pricing options quoted in emerging markets. The market data used to evaluate the option pricing models exhibited some troublesome characteristics common for the developing markets. Although the results suggested that the NN did not outperform the BSM model, analysis of the error pattern indicated that applying additional filtering methods might substantially enhance the performance of the deep learning-based model. Additionally, the ANN price estimations could be incorporated into an ensemble or a stack of different models, e.g., a combination of BSM, Heston, and ANN. Such an approach posed a possible solution to the troublesome data behaviors, which caused the pricing inaccuracies.

Further research for ML methods applied to pricing options on markets, such as the WSE, could focus on dealing with the varying accuracy between moneyness states. As suggested above, either filtering the data or developing models for different conditions could be tested. Secondly, more effort could be put to the detection of outliers and effectively dealing with such observations. Moreover, to provide a more diversified sample, the high-frequency data could be used. Lack of ITM options was the main reason for such biased prices provided by the neural network. Lastly, an additional input data feed (e.g., various moneyness states) for the NN could be employed to provide more detailed information about options diversity.

## Figures and Tables

**Figure 1 entropy-24-00035-f001:**
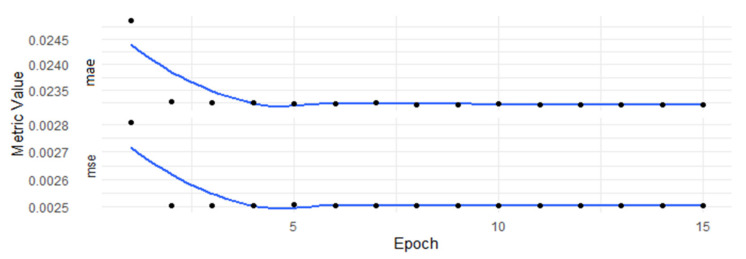
Error metrics estimated during the learning process with respect to the epochs. Values of the MAE and the MSE metrics calculated after every epoch of training the NN with the following hyperparameters: neurons—500; batch size—1000; dropout rate—0.2; optimizer—Adam; activation function—ReLU; learning rate—0.001; β1—0.9; β2—0.9999.

**Figure 2 entropy-24-00035-f002:**
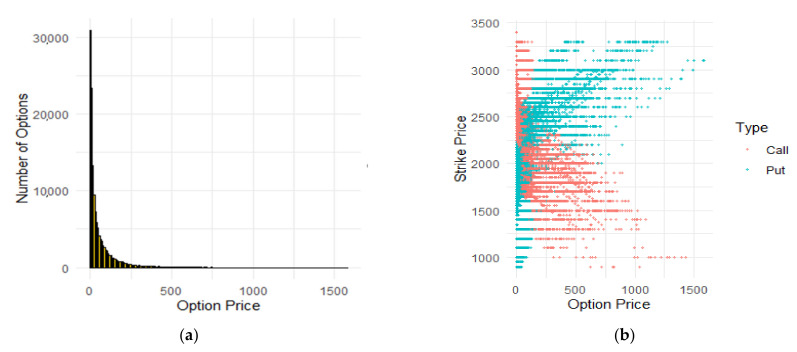
Histograms of the options prices and comparison of market and strike prices. (**a**) The histogram of option prices is strongly influenced by the accumulation of the observations near zero. (**b**) The typical strike prices are between 1800 and 2500.

**Figure 3 entropy-24-00035-f003:**
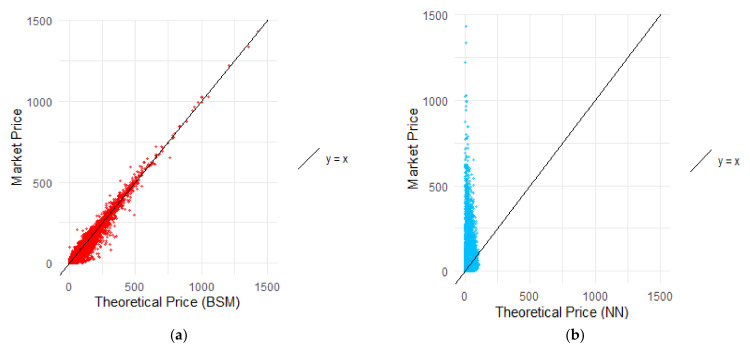
The BSM model prices and market prices along with NN model prices and market prices with curve y = x. The model prices from the BSM model from the out-of-sample data and the model prices from the NN from the out-of-sample data with the corresponding market prices revealing the bias of the models. (**a**) Prices from BSM model. (**b**) Prices from the NN model.

**Table 1 entropy-24-00035-t001:** Possible values of the hyperparameters investigated during the first stage.

Parameter	Options or Range
Neurons (each layer)	250, 500, 1000, 1500, 2500
Batch Size	250, 500, 1000, 1500, 2000, 2500
Epochs	15
Dropout Rate	0, 0.05, 0.1, 0.2
Optimizer	RMSProp, Adam
Activation Function	ELU, ReLU, Softmax, Sigmoid

Note: Values of hyperparameters checked in the first stage of NN architecture development.

**Table 2 entropy-24-00035-t002:** Hyperparameters of the neural network framework.

Parameter	Chosen Option or Value
Neurons (each layer)	1000
Batch Size	1500
Epochs	15
Dropout Rate	0.1
Optimizer	Adam
Activation Function	ReLU

Note: The hyperparameters chosen as the optimal values from all the possibilities in Table 1.

**Table 3 entropy-24-00035-t003:** Values of the hyperparameters investigated during the hyperparameters tuning.

Parameter	Options or Range
Neurons (each layer)	500, 1000, 1500, 2000
Batch Size	1000, 1500, 2000
Dropout Rate	0.05, 0.1, 0.15, 0.2, 0.25
Optimizer	SGD, Adam, Adamax, Adagrad, Adadelta, Nadam
Activation Function	ReLU
Epochs	5
Initializer	Random Normal, Random Uniform, Glorot Normal, Glorot Uniform, Lecun Normal
Learning Rate	0.0001, 0.005, 0.001, 0.005, 0.01
β1	0.75, 0.8, 0.9, 0.95, 0.975
β2	0.95, 0.975, 0.999, 0.9999

Note: Hyperparameters values for the tuning phase aiming to improve the performance of NN.

**Table 4 entropy-24-00035-t004:** Hyperparameters of the neural network framework.

Neurons	MAE	MSE
500	0.0232028	0.0025026
1000	0.0232188	0.0025038
1500	0.0232206	0.0025054
2000	0.0232127	0.0025038

Note: Final values of the error metrics calculated for different numbers of nodes for the hyperparameters tuning. Other hyperparameters: batch size—1500; dropout rate—0.1; optimizer—Adam; activation function—ReLU; learning rate—0.001; initializer—none.

**Table 5 entropy-24-00035-t005:** The batch size and error metrics evaluated after each training process.

Batch Size	MAE	MSE
1000	0.0231536	0.0024729
1500	0.0231796	0.0024737
2000	0.0231300	0.0024713

Note: Final values of the error metrics calculated for different batch sizes for the hyperparameters tuning. Other hyperparameters: neurons—500; dropout rate—0.1; optimizer—Adam; activation function—ReLU; learning rate—0.001; initializer—none.

**Table 6 entropy-24-00035-t006:** The dropout rate and error metrics evaluated after each training process.

Dropout Rate	MAE	MSE
0.05	0.0232059	0.0025036
0.1	0.0232209	0.0025042
0.15	0.0232166	0.0025025
0.2	0.0232186	0.0025025
0.25	0.0232311	0.0025060

Note: Final values of the error metrics calculated for different dropout rates for the hyperparameters tuning. Other hyperparameters: neurons—500; batch size—1000; optimizer—Adam; activation function—ReLU; learning rate—0.001; initializer—none.

**Table 7 entropy-24-00035-t007:** The optimizer and error metrics evaluated after each training process.

Optimizer	MAE	MSE
SGD	0.0235614	0.0025065
Adam	0.0232262	0.0025050
Adamax	0.0232267	0.0025074
Adagrad	0.0232103	0.0025060
Adadelta	0.0237524	0.0025160
Nadam	0.0232326	0.0025073

Note: Final values of the error metrics calculated for different optimizing methods for the hyperparameters tuning. Other hyperparameters: neurons—500; batch size—1000; dropout rate—0.2; activation function—ReLU; learning rate—0.001; initializer—none.

**Table 8 entropy-24-00035-t008:** The initializer and error metrics evaluated after each training process.

Initializer	MAE	MSE
Random Normal	0.0232714	0.0025059
Random Uniform	0.0232260	0.0025062
Glorot Normal	0.0232198	0.0025041
Glorot Uniform	0.0232212	0.0025027
Lecun Normal	0.0232197	0.0025050

Note: Final values of the error metrics calculated for different initializers for the hyperparameters tuning. Other hyperparameters: neurons—500; batch size—1000; dropout rate—0.2; optimizer—Adam; activation function—ReLU; learning rate—0.001.

**Table 9 entropy-24-00035-t009:** The learning rate, β1, β2, and error metrics evaluated after each training process.

Learning Rate	β1	β2	MAE	MSE
0.001	0.9	0.9999	0.0232107	0.0025051
0.005	0.8	0.9999	0.0232112	0.0025075
0.001	0.8	0.9999	0.0232116	0.0025121
0.001	0.8	0.95	0.0232172	0.0025096
0.001	0.9	0.975	0.0232194	0.0025064

Note: Values of error metric for 5 best runs of the NN with corresponding hyperparameters used in the run. Other hyperparameters are: neurons—500, batch size—1000, dropout rate—0.2, optimizer—Adam, activation function—ReLU, initializer—lecun normal.

**Table 10 entropy-24-00035-t010:** Summary statistics for selected variables.

	Mean	Standard Deviation	Minimum	Median	Maximum
Option Price	59.820	101.651	0.01	21.4	1580
WIG20 Index	2265.15	266.968	1327.64	2308.44	2932.62
Strike Price	2252.22	384.034	900	2275	3400
Interest Rate	0.027	0.013	0.016	0.017	0.058
Dividend Rate	0.029	0.011	0.011	0.030	0.060
Time to Maturity	0.352	0.330	0.000	0.250	1.460
Historical Volatility	0.1847	0.073	0.0675	0.1656	0.5087

Note: Statistics calculated for the Polish market in years 2009–2019. All options quoted on the WSE during that period are summarized.

**Table 11 entropy-24-00035-t011:** Summary statistics for options concerning their moneyness.

Type	Moneyness	Number of Options	Mean Price	Minimum Price	Maximum Price
Call	OTM	35,888	14.83	0.01	256.60
	ATM	25,259	59.40	0.01	400.00
	ITM	7138	265.30	30.00	1429.00
Put	OTM	40,771	17.11	0.01	289.00
	ATM	23,219	64.58	0.01	468.00
	ITM	7096	312.00	22.00	1580.1

Note: Summary statistics for all the options quoted on the WSE in years 2009–2019 with distinction between types and moneyness.

**Table 12 entropy-24-00035-t012:** Cross-validation training and testing errors.

	Fold	1	2	3	4	5
Train	MAE	0.023221	0.023141	0.023278	0.023200	0.023331
	MSE	0.002539	0.002448	0.002521	0.002524	0.002496
Test	MAE	0.023335	0.023624	0.023052	0.023298	0.022861
	MSE	0.002418	0.002795	0.002436	0.002441	0.002501

Note: Values of the error metrics calculated after training of the NN on a partial dataset and testing on the data left from the partition. The hyperparameters: neurons—500; batch size—1000; dropout rate—0.2; optimizer—Adam; activation function—ReLU; learning rate—0.001; β1—0.9; β2—0.9999.

**Table 13 entropy-24-00035-t013:** Error metrics for the prices estimated using the BSM model.

Type	Moneyness	MAE	MSE	RMSE	MAPE
Call	OTM	**11.293**	436.624	20.896	1.2445
	ATM	13.470	422.868	20.564	0.3764
	ITM	19.709	766.129	27.679	**0.0913**
Put	OTM	**7.529**	184.029	13.566	0.6277
	ATM	12.779	409.507	20.236	0.2733
	ITM	24.304	1197.690	34.608	**0.0904**

Note: The values of the error metrics for prices obtained using the BSM model, divided between types and moneyness of the options in an in-sample period.

**Table 14 entropy-24-00035-t014:** Error metrics for the prices estimated using the NN model.

Type	Moneyness	MAE	MSE	RMSE	MAPE
Call	OTM	**22.763**	758.021	27.532	16.758
	ATM	39.516	2967.675	54.476	2.3205
	ITM	242.46	78,512.48	280.201	**0.9020**
Put	OTM	**14.626**	374.482	19.351	10.508
	ATM	44.773	3683.993	60.696	1.9122
	ITM	282.897	113,397	336.745	**0.8903**

Note: The values of the error metrics for prices obtained using the BSM model divided between types and moneyness of the options within the in-sample period

**Table 15 entropy-24-00035-t015:** Error metrics for the BSM model prices.

Type	Moneyness	MAE	MSE	RMSE	MAPE
Call	OTM	**11.009**	404.216	20.105	1.234
	ATM	13.835	445.259	21.101	0.3634
	ITM	20.173	839.290	28.970	**0.0915**
Put	OTM	**7.950**	201.196	14.184	0.6317
	ATM	13.047	409.144	20.227	0.2927
	ITM	23.863	1156.281	34.004	**0.0939**

Note: The values of the error metrics for prices obtained using the BSM model divided between types and moneyness of the options in the in-sample period

**Table 16 entropy-24-00035-t016:** Error metrics for the neural network out-of-sample prices.

**Type**	**Moneyness**	**MAE**	**MSE**	**RMSE**	**MAPE**
Call	OTM	**22.993**	767.184	27.698	17.124
	ATM	39.781	2937.782	54.201	2.135
	ITM	246.013	80,779.38	284.217	**0.9041**
Put	OTM	**14.742**	381.378	19.529	10.505
	ATM	44.745	3670.09	60.581	1.974
	ITM	277.419	107,795.5	328.322	**0.8897**

Note: The values of the error metrics divided between types and moneyness of the options prices obtained using the NN with the following hyperparameters: neurons—500; batch size—1000; dropout rate—0.2; optimizer—Adam; activation function—ReLU; learning rate—0.001; β1—0.9; β2—0.9999.

## Data Availability

All relevant data are within the paper.

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
