# Peer review of "Artificial Neural Networks Performance in WIG20 Index Options Pricing"

_entropy, 2021, doi:10.3390/e24010035_

Round 1
Reviewer 1 Report
Please see attached review report.

Author Response
The detailed replies to the Reviewer are in the attached .pdf document.

Reviewer 2 Report
The primary focus of the artilce is the performance comparison of Neural Networks and Black-Scholes –Merton models in option pricing. The article brings new scientific findings regarding the methodological modelling of options. Link to Entropy is not clear to me from the title. Maybe this should be further emphasized (the article probably related to compelx systems, networks or computing as it deals with information loss too). The article contains a short discussion and it really compares the results with international findings. The abstract clearly represent the article content. The aim and methodology is clearly specified but the main results should be corrected. I don’t see how the topic fits in the scope of the journal’s scope, this should be clearified. The introduction is brief and contains the research gap which is stated clearly. The theoretical part of the article is all rigth, it contains 48 sources (mainly from databases such as Scopus and Web of Science). 63% of the sources are not older than 15 years. I found too many old references, I think these should be decresed. Materials and Methods section is too long and unsystematicly structured (see from lines 265 – 354). Figures 1 should be replaced elsewhere too. There are too many mistakes (grammar mostly) which can be seen from the corrected pdf version which I will provide for the authors. I find the article very interesting but the writing is proof. The tenses should be corrected. I actually proofread the article but authors should find a native speaker to check my suggestions too. The methodology part should be shortened. The models were described in a clear way. There were several scientific hypotheses formed and evaluated in a proper way. The conclusion is perfect, it is written in an interesting way and the direction of the future research is also there so as the limitation of the study (Polish market characteristics). The article satisfies the formal requirements. I think the rerecences and the template is according to the journal requirements. All the mistakes were indicated in the PDF file and will be available for the authors. I see hugh potentials in the article and I would suggest a minor revision just because of the grammer mistakes and not because of the professional quality. To sum it up, the article’s structure should be changed a little bit and should be corrected grammarly.
Some minor issues (apart from the so many grammar mistakes):
ML should be intorduced in line 36
In line 150 P_C and P_P should be corrected
In line 214 change the wording „Thanks to”
Avoid using „we” like in line 258 and elsewhere
3.3.3. Chapter’s title is the same as 3.3.2. in line 223.
Table 4-9 should be in the result section
Figure 1 should be placed elsewhere as indicated in the corrected PDF
In line 415 the applied transformation is not a scaling but a scaling and centering which is called normalization. the mean of the X should be denoted otherwise in Formula (13).
I thin „more accurate” is related to RMSE and „lowest percentage bias” is related to MAPE. You should carefully check the result from this point of view

Author Response

(The authors gave the same response as above.)

Reviewer 3 Report
The paper gives a thorough comparison of the performance of artificial neural networks (ANNs) in option pricing with the results obtained from the Black–Scholes–Merton model. Different error metrics and the components of ANNs are given in the manuscript. The authors provide real data analysis based on data from the Warsaw Stock Exchange. The paper appears to be well-written and comprehensively referenced.
Author Response

(The authors gave the same response as above.)
